# Resting-State EEG Connectivity at High-Frequency Bands and Attentional Performance Dysfunction in Stabilized Schizophrenia Patients

**DOI:** 10.3390/medicina59040737

**Published:** 2023-04-09

**Authors:** Ta-Chuan Yeh, Cathy Chia-Yu Huang, Yong-An Chung, Sonya Youngju Park, Jooyeon Jamie Im, Yen-Yue Lin, Chin-Chao Ma, Nian-Sheng Tzeng, Hsin-An Chang

**Affiliations:** 1Department of Psychiatry, Tri-Service General Hospital, National Defense Medical Center, Taipei 114202, Taiwan; 2Department of Life Sciences, National Central University, Taoyuan 320317, Taiwan; 3Department of Nuclear Medicine, College of Medicine, The Catholic University of Korea, Seoul 07345, Republic of Korea; 4Department of Psychology, Seoul National University, Seoul 08826, Republic of Korea; 5Department of Emergency Medicine, Tri-Service General Hospital, National Defense Medical Center, Taipei 114202, Taiwan; 6Department of Emergency Medicine, Taoyuan Armed Forces General Hospital, Taoyuan 325208, Taiwan; 7Department of Psychiatry, Tri-Service General Hospital Beitou Branch, National Defense Medical Center, Taipei 112003, Taiwan

**Keywords:** schizophrenia, attentional dysfunction, cognitive deficits, electroencephalography, functional connectivity

## Abstract

*Background and Objectives*: Attentional dysfunction has long been viewed as one of the fundamental underlying cognitive deficits in schizophrenia. There is an urgent need to understand its neural underpinning and develop effective treatments. In the process of attention, neural oscillation has a central role in filtering information and allocating resources to either stimulus-driven or goal-relevant objects. Here, we asked if resting-state EEG connectivity correlated with attentional performance in schizophrenia patients. *Materials and Methods*: Resting-state EEG recordings were obtained from 72 stabilized patients with schizophrenia. Lagged phase synchronization (LPS) was used to measure whole-brain source-based functional connectivity between 84 intra-cortical current sources determined by eLORETA (exact low-resolution brain electromagnetic tomography) for five frequencies. The Conners’ Continuous Performance Test-II (CPT-II) was administered for evaluating attentional performance. Linear regression with a non-parametric permutation randomization procedure was used to examine the correlations between the whole-brain functional connectivity and the CPT-II measures. *Results*: Greater beta-band right hemispheric fusiform gyrus (FG)-lingual gyrus (LG) functional connectivity predicted higher CPT-II variability scores (r = 0.44, *p* < 0.05, corrected), accounting for 19.5% of variance in the CPT-II VAR score. Greater gamma-band right hemispheric functional connectivity between the cuneus (Cu) and transverse temporal gyrus (TTG) and between Cu and the superior temporal gyrus (STG) predicted higher CPT-II hit reaction time (HRT) scores (both r = 0.50, *p* < 0.05, corrected), accounting for 24.6% and 25.1% of variance in the CPT-II HRT score, respectively. Greater gamma-band right hemispheric Cu-TTG functional connectivity predicted higher CPT-II HRT standard error (HRTSE) scores (r = 0.54, *p* < 0.05, corrected), accounting for 28.7% of variance in the CPT-II HRTSE score. *Conclusions*: Our study indicated that increased right hemispheric resting-state EEG functional connectivity at high frequencies was correlated with poorer focused attention in schizophrenia patients. If replicated, novel approaches to modulate these networks may yield selective, potent interventions for improving attention deficits in schizophrenia.

## 1. Introduction

Schizophrenia is a debilitating mental illness, characterized by positive symptoms, negative symptoms and cognitive impairments. Neurocognitive deficits are core symptoms of this mental illness, and their profiles include impairment in attention, learning, working memory, executive function and processing speed. Atypical antipsychotics, as the mainstay of pharmacological treatment for schizophrenia, have little beneficial effect on neurocognitive impairments [1]. Thus, cognitive deficits remain one of the most critical unmet therapeutic needs in schizophrenia. In-depth investigations into the neural underpinning of these deficits are needed to foster the development of effective treatment strategies.

The impairment in attention as an endophenotype of schizophrenia [2] as well as one of the complaints of patients’ everyday life has been the focus of extensive research on this disorder. In addition, research indicates that intact attentional functioning is one of the prerequisites for higher-order cognitive functions and correlates with the ability of schizophrenia patients to live independently. Attentional dysfunction can largely contribute to functional impairment and poor health-related quality of life ubiquitous to schizophrenia [3,4]. The continuous performance test (CPT) is one of the most widely used instruments for assessing attention in patients with schizophrenia [5]. Most studies on CPT in patients with schizophrenia have revealed deficits in attentional performance (i.e., attention/vigilance) that are independent of the clinical state (i.e., being observed not only during the acute phase but also during the stabilization phase of the disease) and can be heritable and modified by schizophrenia susceptibility gene variants [6,7,8]. Specifically, a previous study measured sustained attention by using CPT in first-episode schizophrenia patients, their non-psychotic first-degree relatives and healthy controls [2]. Compared to healthy individuals, patients showed worse performance on all measures of CPT and their non-psychotic first-degree relatives were also impaired on hit reaction time, a CPT index reflecting psychomotor processing speed of the correct response. The study results suggested the use of the hit reaction time measure of the CPT as an endophenotype marker for schizophrenia. Diverse structural and/or functional neurobiological alterations in schizophrenia are involved in different mechanisms of attentional deficits, giving rise to different patterns of poor performance in CPT. The standard Conners’ CPT (CPT-II) is a computer-administered test that is designed to measure a person’s sustained and selective attention. The CPT-II takes fourteen minutes to administer, excluding the recommended practice test, and is a frequently used modality in clinical evaluations. It encompasses a variety of measures, capturing different dimensions of attention and detecting a range of alterations in attentional subfunctions (e.g., focused attention, impulsivity, sustained attention and vigilance) and its reliability has been established in patients with schizophrenia (i.e., the intra-class correlation coefficients for CPT-II indexes ranging from 0.66 to 0.79) [5,9].

There is convergent evidence that aberrant neural oscillations at low- and high-frequencies constitute an important aspect of the failure to generate coherent cognition, causing the characteristic cognitive deficits of schizophrenia [10,11]. These deficits are attributed to N-methyl-D-aspartate receptor hypofunction and could be assessed in schizophrenia patients with electroencephalography (EEG) [12]. There is a growing body of research examining EEG characterization as a neurophysiological biomarker of attentional dysfunction in schizophrenia and aiming to identify the neural substrates of attentional dysfunction as targets for developing therapeutics [13]. For example, a recent study indicated the EEG (electroencephalogram) biomarkers for attentional functioning in schizophrenia patients by investigating event-related potential (ERP) and source localization analyses of EEG recordings during CPT [14]. Schizophrenia is known as a mental illness of disturbed communication across neural networks [15]. Research has indicated altered functional connectivity of brain networks in patients with schizophrenia [16]. In these patients, disturbed functional connectivity is further assumed to underlie their neurocognitive deficits. Investigation into the neural networks involved in this core feature of schizophrenia would help to clarify the pathophysiological mechanisms underlying the emergence of cognitive deficits (e.g., attentional dysfunction, which is more relevant for this study). Earlier research on connectivity in schizophrenia focused on task-related connectivity metrics. Recent research has noticed the importance of resting-state, spontaneous activity of the brain in the processing of incoming stimuli [17] and the underpinnings of cognitive disturbances [18,19]. For example, EEG gamma synchrony abnormalities play an important role in affecting functional connectivity in central executive circuits and causing neurocognitive symptoms in schizophrenia [10]. A recent study indicated that patients with first-onset schizophrenia had a reduced ability to modulate task-evoked changes in EEG gamma synchrony in the context of higher resting-state synchrony and that these gamma synchrony abnormalities were associated with performance on the CPT [20].

Resting-state EEG can characterize brain network properties by investigating spatiotemporal patterns of neuronal activity in many different frequencies with excellent temporal resolution [21]. The mechanism of synchronous neural oscillations underpins the connectivity within large-scale whole brain networks. Accumulating evidence indicates that the coupling patterns of brain networks can be well captured by certain connectivity metrics of resting-state EEG based on phase correlations among signals [18,21]. Research has reported abnormally organized resting-state source-level EEG connectivity in schizophrenia, with results indicating reduced phase synchronization in the alpha frequency and increased phase synchronization in the delta, theta, beta and gamma frequency ranges [16]. It is believed that disturbed resting-state functional connectivity of brain networks is responsible for the characteristic neurocognitive deficits in patients with schizophrenia [10]. For example, gamma-band oscillations are involved in both local and large-scale neuronal synchronization, underlying a broad range of higher-order cognitive functions typically impaired in schizophrenia. Research indicates that abnormalities in EEG gamma-band synchrony (e.g., reduced stimulus-evoked gamma-band oscillations coupled with resting-state cortical gamma-band hyperconnectivity) play a vital role in the neural mechanisms associated with cognitive deficits [22]. Furthermore, theta-band oscillations are crucial for the long-range neural synchronization of cortical networks. Recent research indicated that resting-state EEG theta-band hyperconnections between the posterior cingulate cortex, cuneus and precuneus impeded timely initiation of cognitive performance in patients with first-episode schizophrenia [23]. Glutamate N-methyl-D-aspartate receptor (NMDAR) dysfunction and reduced glutamatergic signaling coupled with downstream GABAergic deficits have been considered as a potential mechanism underlying the aberrant resting-state EEG gamma- and theta-band connectivity in schizophrenia [22,24].

Since attentional performance is crucial for functional outcomes in patients with schizophrenia, clinicians have to monitor and manage sustained attention deficits in these patients. The CPT-II is a commonly used instrument by clinical assessments for detecting the degree of attentional deficits in patients with schizophrenia. As far as we know, there is still a lack of research elucidating the relationships between the organizations of resting-state EEG functional connectivity and attentional performance dysfunction during higher-order cognitive processing in the CPT-II. To fill the gap in the available literature, the primary aim of the current study was to examine the correlations between attentional performance assessed by CPT-II and resting-state EEG whole-brain source-based source-level connectivity across a wide range of frequencies in stabilized schizophrenia patients. As secondary aims, we investigated whether EEG connectivity was correlated with the results of clinical measures and other cognitive tests.

## 2. Materials and Methods

### 2.1. Participants

The study is a secondary analysis of the pooled baseline data from 2 published double-blind, randomized, sham-controlled trials (RCTs) investigating the effects of transcranial electrical stimulation in schizophrenia patients approved by the Institutional Review Board of the Tri-Service General Hospital, Taiwan; written informed consent was obtained from all patients prior to the study onset, and full details of the protocol were reported elsewhere [25,26]. The analysis focused on the pooled baseline data of the cognitive function test (CPT) and resting-state EEG from all the participants. As a summary in Table 1, there were two samples from the two independent RCTs. The inclusion criteria of sample 1 were (1) aged 20–65, (2) fulfilling DSM-5 diagnosis of schizophrenia or schizoaffective disorder, (3) illness duration ≥1 year, (4) being treated with an adequate and stable therapeutic dose of antipsychotics for at least 8 weeks before enrollment, (5) a Positive and Negative Syndrome Scale (PANSS) score of 120 points or less at both screening and baseline. The exclusion criteria were (1) having any active substance use disorder (except for caffeine and/or tobacco) or current psychiatric comorbidity, (2) having implanted metal or cerebral medical devices in the head, (3) being a pregnant or lactating woman and (4) having any history of cerebrovascular diseases, seizures, intracranial neoplasms/surgery, or severe head injuries. The inclusion criteria of sample 2 were (1) patients aged 20–65 with DSM-5-defined schizophrenia or schizoaffective disorder, (2) duration of illness > 2 years and (3) being clinically stable and on an adequate therapeutic dose of antipsychotics for at least 8 weeks before enrolment. The exclusion criteria were essentially identical to those for sample 1.

### 2.2. Continuous Performance Test and Other Cognitive Tests

Cognitive performance was assessed using standardized tests administered by a well-trained examiner. All participants were instructed to abstain from caffeine for at least 24 h before the assessments. Our main cognitive outcome measure was attentional performance, as assessed by the Continuous Performance Test, 2nd Edition (CPT-II) [5]. During the 14 min CPT-II, participants were asked to press the space bar on a conventional keyboard every time any letter showed up on a computer screen (a total of 324 times), but not to press the space bar when the letter “X” showed up (a total of 36 times). Each letter was shown for 250 ms. CPT-II had six blocks and each block was composed of three parts with twenty elements each. The parts in each block had an inter-stimulus interval (ISI) of 1, 2, or 4 s, with the order altered. CPT-II items of hit reaction time (HRT), perseveration (PER) and commission errors (COM) measured impulsivity. Detection (d’), variability of standard error (VAR), omission errors (OM) and HRT standard error (HRTSE) measured focused attention. HRT block change (HRTBC) measured sustained attention. HRT interstimulus interval changes (HRTISIC) measured vigilance.

In addition, other cognitive outcome measures were assessed by a variety of cognitive tests. Color Trails Test (CTT) was used to assess attention and visuomotor processing speed, attentional set-shifting, executive function and visuospatial working memory [27]. During the Wisconsin Card Sorting Test (WCST), all participants had to sort sixty-four cards based on the numbers, shapes and colors by applying their abstract reasoning ability and cognitive flexibility in response to changing environmental contingencies [28]. Perseverative errors occurred when participants persisted in using the wrong rule despite negative feedback. WCST perseverative errors were used to measure executive function. The Tower of London (TOL)-Drexel University Test 2nd Edition (TOL^DXtm^) was administered to assess planning, processing and problem-solving skills [29]. TOL accuracy indicates the total correct score in the TOL. TOL time indicates the total time spent in the TOL. The Stroop interference test on the Vienna test system, a computerized color-word interference paradigm by Stroop [30], was used to assess selective attention and cognitive flexibility. The test is based on the assumption that the reading speed of a color-word is slower if the word is written in a differently colored font. There is always a delay in naming the color of this word if the color and color-word do not match. During the task, all participants had to press the correct respective entry field or color button as fast as possible. The color-word interference tendency (i.e., impairment in the reading speed or color recognition due to interfering information) was registered.

### 2.3. Clinical Measures

The rating tool of Positive and Negative Syndrome Scale (PANSS) was used to measure the severity of 30 psychopathological symptoms of schizophrenia, divided into positive, negative and general psychiatric symptoms. The Personal and Social Performance scale (PSP) was used to assess different domains of psychosocial function including socially useful activities, personal and social relationships, self-care, and disturbing and aggressive behaviors. The summary instruction table of the PSP scale defines the final global score, ranging from 1 to 100, with a higher score indicating better psychosocial function. As reported in our prior work [25,26], all clinical assessments were performed by a well-trained and experienced research psychiatrist (H.C.) and the intraclass correlation coefficient for the intra-rater repeatability was between excellent and good (0.92 for the PANSS and 0.89 for the PSP). The self-reported Beck Cognitive Insight Scale (BCIS) was used to assess patients’ cognitive insight. The scale contains 15 items, which are divided into 2 subscales of reflective attitude (9 items) and certain attitude (6 items). The reflective attitude subscale score minus the certain attitude subscale score yields the R-C index, which indicates the level of cognitive insight. Higher R-C index scores represent better cognitive insight. The participants’ daily dosage of antipsychotic drugs equivalent to 100 mg chlorpromazine was calculated.

### 2.4. Estimations for EEG Source Localization

The methods can be found in our previous publications [31,32], but the procedures are outlined here for completeness. Participants were instructed not to drink coffee, tea, or any other stimulant-containing beverage 1 h before and alcohol 24 h before the beginning of the recording session. The recordings took place in an electrically shielded and light and sound-attenuated room. Resting-state EEG (rsEEG) was collected at a sampling rate of 4000 Hz with 32 Ag/AgCl sintered ring electrodes mounted on an elastic cap (NP32, GmbH, Ilmenau, Germany) according to the International 10–20 system, using Neuro Prax^®^ TMS/tES compatible full band DC-EEG system (NeuroConn GmbH, Ilmenau, Germany) with a hardware EEG band-pass filter (0–1200 Hz). The recording channels were connected to a reference electrode located at the tip of the nose and the ground electrode was set in the Fpz position. The impedance value of the electrodes was kept below 5 kΩ. Eye movements and blink artifacts were recorded with 4 electrooculogram (EOG) channels. The Neuro Prax^®^ EEG system built-in software processed and automatically corrected EEG artifacts from eye movements, blinks and small body movements. rsEEG was continuously recorded for a total of 10 min while the participants were at rest with closed eyes (5 min) and opened eyes (5 min). The participants’ sequence of the eye conditions was counterbalanced and randomized. During the eyes-open condition, participants were instructed to visually fixate on a small crosshair presented on a computer screen in front of them. They were instructed to relax while staying awake with eyes closed during the eyes-closed condition. Offline, EEG data were exported to EDF from the native format and analyzed in the EEGLAB v2020.0 [33]. Transient or large-amplitude artifacts in EEG recordings were detected and rejected by using the artifact subspace reconstruction (ASR) plug-in of EEGLab [34]. Independent component analysis (ICA) decomposed EEG signals into a series of spatially fixed and temporally independent components (ICs). IClabel, an automatic EEG IC classification plug-in for EEGLAB [35], was used to distinguish ICs as brain or non-brain sources. ICs corresponding to non-cerebral artifactual sources were removed. Only accepted epochs of eyes-closed rsEEG data were chosen for EEG source localization.

The source-localized signals were estimated using the exact low-resolution brain electro-magnetic tomography (eLORETA) software, a three-dimensionally (3D) distributed linear inverse solution, allowing zero localization error under noise-free conditions in dealing with the scalp-recorded EEG signals [36]. eLORETA uses a realistic head model [37] based on the MNI152 template [38], with the 3D solution space restricted to cortical gray matter and hippocampi, as determined by the probabilistic Talairach atlas [39]. The intracerebral volume (eLORETA inverse solution space) is partitioned in 6239 voxels at 5 mm spatial resolution (i.e., cubic elements of 5 × 5 × 5 mm).

### 2.5. Whole-Brain Electrical Source-Based Functional Connectivity

eLORETA software was implemented for the analysis of whole-brain functional connectivity from EEG source-based signals. Brodmann areas (BAs) were defined as anatomical labels using the neuroanatomic Montreal Neurological Institute (MNI) space with correction to Talairach space [40]. Regions of interest (ROIs) in eLORETA were created by including a single voxel of gray matter at the centroid of 42 BAs (i.e., BAs: 1-11, 13, 17-25, 27-47) for each hemisphere. The MNI coordinates for the ROIs are listed in Table 2. To estimate the whole-brain functional connectivity, lagged phase synchronization (LPS) between any pair of ROI was calculated to obtain the functional connectivity values of all possible pairs of 84 ROIs (i.e., 84 × 83/2 = 3486 pairs of sources distributed throughout the whole cerebral cortex). Based on normalized Fourier transforms, the LPS algorithm evaluates the similarity between signals in the frequency domain. Hence, LPS is a method associated with nonlinear functional connectivity and represents the connectivity of two signals after excluding the instantaneous zero-lag component of phase synchronization caused by intrinsic artifacts or non-physiological components artifacts (e.g., volume conduction) [36]. Finally, LPS values were obtained for 3486 pairs of ROIs, for each participant and for each of the five independent EEG frequency bands of delta (1–3.5 Hz), theta (4–7.5 Hz), alpha (8–12.5 Hz), beta (13–32.5 Hz) and gamma (33–45 Hz).

### 2.6. Statistical Analyses

SPSS Statistics 21.0 software (IBM SPSS Inc., Chicago, IL, USA) and eLORETA-based statistical non-parametric maps (SnPM) [36] were used for statistical analyses. In the SnPM analysis tool, linear regression was used to examine the significant correlations between the whole-brain electrical source-based functional connectivity and the CPT-II measures or other measures. For each analysis, tests were performed to examine all connections (i.e., LPS) between 84 ROIs across the five frequency bands (3486 × 5). The eLORETA implements a non-parametric permutation randomization procedure based on “maximal statistic” to correct for multiple comparisons associated with multiple ROIs and frequency bands. The omnibus null hypothesis was rejected if at least one value of the correlation coefficient was above the critical threshold r_crit_ for *p* = 0.05 (corrected) determined by 5000 data randomizations. Statistical significance for the results was set at *p* < 0.05 (two-tailed).

## 3. Results

### 3.1. Sociodemographic and Clinical Characteristics of the Participants

Table 1 shows the variables of demographic and clinical characteristics of the participants. Among a total of 72 patients, 33 were women (45.8%). The mean age was 42.90 ± 10.87 years. The mean mental illness duration was 17.44 ± 10.97 years.

### 3.2. Correlation Analyses

Significant associations were found between scores of certain CPT-II measure items (i.e., VAR, HRT and HRTSE) and whole-brain electrical source-based functional connectivity at high frequency only (i.e., beta and gamma frequency). Whole-brain electrical source-based functional connectivity was not associated with the scores of other CPT-II measure items, PANSS, PSP, BCIS, CTT1, CTT2, WCST non-perseverative error, TOL accuracy, TOL time, Stroop interference test, the daily dose of antipsychotic medications (chlorpromazine equivalents) and illness duration (all *p* values > 0.05).

#### 3.2.1. Correlations between CPT-II VAR Score and EEG Source-Based Functional Connectivity

The VAR score (variability of standard error) indicates “within respondent” variability, i.e., the amount of variability the individual shows in relation to his or her own overall standard error. The VAR score represents variability in attention, or consistency in hit reaction time, over time. A high value indicates a great inconsistency in the response speed, often related to inattentiveness. As can be seen in Figure 1A,B, the CPT-II VAR score was significantly associated with the beta-band (13–32.5 Hz) LPS between the right hemispheric fusiform gyrus (FG) and lingual gyrus (LG). The right hemispheric FG-LG LPS accounted for 19.5% of variance in the CPT-II VAR score. The patients with higher CPT-II VAR scores had greater beta-band EEG right hemispheric FG-LG functional connectivity (r = 0.44, *p* < 0.05, corrected, Figure 1C). The inclusion of the PANSS total score and antipsychotic exposure in chlorpromazine dose equivalents did not substantially change the results.

#### 3.2.2. Correlations between CPT-II HRT Score and EEG Source-Based Functional Connectivity

HRT score indicates the mean response time (ms) for all target responses over all six trial blocks of CPT-II. It can assess response speed consistency. HRT averaging over 900 ms is considered a sluggish response, often related to inattentiveness. As can be seen in Figure 2A, the CPT-II HRT score was significantly associated with the gamma-band (33–45 Hz) LPS between the right hemispheric cuneus (Cu) and transverse temporal gyrus (TTG) and between the right hemispheric Cu and superior temporal gyrus (STG). The right hemispheric Cu-TTB and Cu-STG LPS accounted for 24.6% and 25.1% of variance in the CPT-II HRT score, respectively. The patients with higher CPT-II HRT scores had greater gamma-band EEG right hemispheric functional connectivity between Cu and TTG (r = 0.50, *p* < 0.05, corrected, Figure 2B) and between Cu and STG (r = 0.50, *p* < 0.05, corrected, Figure 2C). The inclusion of the PANSS total score and antipsychotic exposure in chlorpromazine dose equivalents did not substantially change the results.

#### 3.2.3. Correlations between CPT-II HRTSE Score and EEG Source-Based Functional Connectivity

The HRTSE score reflects the consistency of response time on CPT-II. A high value of HRTSE indicates a high variability in HRT, often related to inattentiveness. As can be seen in Figure 3A,B, the CPT-II HRTSE score was significantly associated with the gamma-band (33–45 Hz) LPS between the right hemispheric Cu and TTG. The right hemispheric Cu-TTG LPS accounted for 28.7% of variance in the CPT-II HRTSE score. The patients with higher CPT-II HRTSE scores had greater gamma-band EEG right hemispheric functional connectivity between Cu and TTG (r = 0.54, *p* < 0.05, corrected, Figure 3C). The inclusion of the PANSS total score and antipsychotic exposure in chlorpromazine dose equivalents did not substantially change the results.

## 4. Discussion

Considering that attentional deficits are associated with functional outcomes in schizophrenia, the main goal of this study was to identify the relationship between the attentional performance dysfunction during higher-order cognitive processing in the CPT-II and the arrangement of resting-state neuronal synchronizations in a group of stabilized schizophrenia patients and to identify the neurophysiological correlates of described deficits. The rationale for such an attempt is that elucidating the neurophysiological underpinning of the attentional deficits is potentially the first step to developing novel therapeutic strategies. To the best of our knowledge, there is still a lack of research that demonstrates the relationships between the organizations of resting-state functional connectivity and attentional performance on the CPT-II; therefore, we have not limited the analysis to specific brain regions of interest or specific frequency bands. Of note, our study analyzed resting-state recordings of the source activity for EEG based on the following premise: the altered resting-state functional connectivity may have a reasonable link with a given type of cognitive deficit because the structure of intrinsic neural synchronizations can reflect the brain functions expressed during an active performance of cognitive tasks to a certain extent [41,42]. There were two key findings: i) in our stabilized patients with schizophrenia, a set of cortical regions (i.e., right hemispheric cuneus, superior temporal gyrus and transverse temporal gyrus) that show coordinated activation in the resting state at gamma frequency were associated with the performance of focused attention on the CPT-II, and ii) the beta-band strengthened coupling between sources located within the right hemispheric FG and LG turned out to be a significant predictor of diminished performance of focused attention on the CPT-II.

A recent study in young patients with first-onset schizophrenia investigated scalp-level EEG gamma synchrony for regions associated with the frontoparietal attention and visual networks, at baseline (pre-stimulus or intrinsic) and during CPT-Identical Pairs (CPT-IP), a modified version of CPT using a lower signal–noise ratio, placing greater demand on working memory, and requiring a high response rate than CPT-II [20]. The authors found that schizophrenia patients had higher intrinsic (baseline) gamma synchrony, particularly among the electrodes of frontal regions, but had lower attention task-evoked changes in gamma synchrony. It was also found that these gamma synchrony abnormalities were related to performance on the CPT-IP in a complex, non-linear way. The results of the study indicated that an overall excess of intrinsic (resting-state) gamma synchrony that was unrelated to demands on attention could constrain the attention network from adequately engaging gamma synchrony connectivity during attention processing, supporting the theoretical accounts of gamma synchrony as a core abnormality underpinning cognitive deficits in schizophrenia. However, this study is limited to the scalp-level analyses in drawing inferences regarding the underlying sources of electrical activity in the brain because of known methodological limitations associated with volume conduction and reference electrode placement. So far, a few studies have assessed EEG connectivity at the source level in schizophrenia and reported increased connectivity in the gamma-band range. However, none of them investigated the relationships between source-level gamma-band connectivity and the attentional performance on CPT-II.

Our study indicated that patients with a diminished performance of focused attention had greater right hemispheric functional connectivity between Cu and TTG and between Cu and STG at gamma frequency. The cuneus (Brodmann area 17) is located in the medial occipital gyri and has a crucial role in both primary and secondary visual processing. It receives input from the contralateral superior retina corresponding to the lower visual field and transmits information to the inferior temporal cortex via the ventral stream, which is involved in form recognition and object representation, and to the posterior parietal cortex via the dorsal stream, which is involved in the representation of object locations and spatial awareness. The anterior part of the cuneus is involved in the neural mechanisms necessary for visual attention, a cognitive process that mediates the selection of important information from the environment. A previous study investigated regional cerebral blood flow (CBF) changes during a sustained attention task in schizophrenia and found that the right cuneus of the patients was increased during the task [43]. A recent study indicated that the cuneus provided an accessible pathway through which the parietal and prefrontal regions can interconnect, and a greater attention task-evoked activation in the cuneus was associated with better attentional performance, indicating a greater ability to allocate attentional resources [44]. Research has also found that schizophrenia patients are characterized by increased connectivity of the cuneus in large-scale brain networks [45].

The transverse temporal gyrus (also called Heschl’s gyrus, occupying Brodmann areas 41 and 42) and the superior temporal gyrus (containing Brodmann areas 41 and 42, and Wernicke’s area, Brodmann area 22) were the primary auditory regions and were also known as one generator of mismatch negativity (MMN), a specific event-related potential (ERP) considered to represent a prediction error process in the brain, a component of the chain of brain events causing attention switches to changes in the environment as well as being a biomarker for schizophrenia [46]. Recent research indicated that impairments in MMN reflected impaired functioning of a broadly distributed brain network and contributed to deficits in both auditory and visual domains of cognitive functioning [47]. Previous research indicated that schizophrenia patients had reduced cortical gray matter thickness of the right superior temporal gyrus (incorporating transverse temporal). Furthermore, a schizophrenia-specific (i.e., being absent in healthy controls) relationship between thinning in the right STG/TTG and poorer attention performance on the continuous performance test was identified [48]. A recent study reported that schizophrenia patients had thinner cortices within temporal regions and that there was a schizophrenia-specific, robust relationship between right transverse temporal thickness and attention performance assessed by the Measurement and Treatment Research to Improve Cognition (MATRICS) consensus cognitive battery [49].

Increased resting gamma-band connectivity between these regions in our patients with schizophrenia possibly reflected compensatory processes for anatomical/functional deficits in these regions. Parts of the task-negative cortical networks in our patients with schizophrenia may be in a state of neural ‘over-connection’ prior to engaging in high-level stimulus processing of CPT-II, involving connections between sensory (occipital) and (temporal) cortices. They were then unable to increase their gamma synchrony in the task-positive cortical networks (i.e., frontoparietal network) in response to an attention-demanding task and sufficiently recruit task-related processes.

EEG beta rhythm oscillations play an important role in top-down signaling and feedback influences for cortical neurons that effectively support the precise synchronization of neuronal discharges [11,50]. Patients with schizophrenia were reported to have disturbances in both top-down signaling and EEG beta-band coordination [50]. Previous research indicated that resting-state EEG beta power was correlated with different cognitive functions including attention and vigilance [50]. Since the beta rhythm-dependent control gain of sensory signals is essential for attention and vigilant performance of the tasks requiring continuous monitoring of stimulus streams and appropriate responses to cued/salient stimuli, insufficient beta-band power may contribute to deficits in these attention functions in schizophrenia patients [50]. Furthermore, recent research suggests that schizophrenia patients are characterized by altered EEG long-range temporal correlations (i.e., measures indicative of neuronal networks operating in a critical state) at beta-band frequency [10]. All these findings point to the importance of EEG beta-band oscillations and connectivity in the attentional performance of schizophrenia patients.

Our results indicated a correlation between increased beta-band right hemispheric FG-LG connectivity and diminished performance of focused attention on the CPT-II. The lingual gyrus (LG, also called medial occipitotemporal gyrus) is involved in processing vision (particularly related to letters), encoding visual memories and analyzing logical conditions. The fusiform gyrus (FG, also called occipitotemporal cortex, Brodmann area 37) is involved in high-level visual processing, specifically object recognition and category identification, of complex stimuli such as faces. Both FC and LG are extra-striate visual cortices and are also considered part of the ventral stream as described earlier. A reduction in FG volume, a compensatory increase in FG activity during tasks and an increase in right FG degree centrality (i.e., functional connectivity hub architecture) both at rest and during N-back tasks have been reported in patients with schizophrenia [51,52,53,54]. A recent study indicated that the grey matter volume of right FG was positively correlated with the performance in cognitive tests in patients with schizophrenia [55]. On the other hand, earlier research indicated that patients with schizophrenia had a reduction in cortical gyrification (or folding) involving right LG and an increase in bilateral LG degree centrality during N-back tasks [54]. Accumulating evidence indicated hyperactivity in bilateral LG and increased connectivity between the LG and other brain regions in schizophrenia patients [56,57]. Taken together, the result of increased resting-state beta-band FG-LG connectivity in the right hemispheric for predicting poorer attentional performance may suggest a role of the conjoint dysfunction of these two regions in the pathophysiology of attentional deficits in schizophrenia.

Our study has some limitations. First, the results of resting EEG source-level large-scale network connectivity in our schizophrenia patients should be interpreted cautiously because we had no control group of healthy individuals and because there was some heterogeneity among the two samples analyzed. Further studies enrolling both homogeneous patients and healthy individuals are needed to confirm our results. Second, we cannot exclude the correlations between the connectivity of other important large-scale networks and attentional performance on CPT-II in schizophrenia patients because eLORETA has the intrinsic limitation of seed-based connectivity between pre-defined brain regions. Third, although previous research has shown that EEG high-frequency EEG activity is independent of antipsychotic medication status [58] and it seems unlikely that equivalent doses of antipsychotics have direct effects on our results, we cannot completely exclude the possibility of indirect or nonlinear effects of antipsychotic drugs on the results. Fourth, the quality of our EEG signals can be further improved by using novel denoising methods, e.g., multiscale principal component analysis (MSPCA) which combines non-centered PCA on approximations and details in the wavelet domain and a final PCA [59,60,61,62]. Specifically, empirical wavelet transform decomposes denoised EEG signals into different modes and the two-dimensional modeling of these modes is applied to identify the variations in different signals. Next, a single geometrical feature is extracted from the 2D modeling of modes and the extracted feature vectors are provided to the machine learning and feedforward neural network (e.g., convolutional neural network, CNN) classifiers and they cascade forward neural networks for a classification check [62]. Recent research identified the underlying patterns of EEG signals of a seizure and a psychiatric phenotype (e.g., depression) by using a novel approach based on geometric features derived from the EEG signal shape of the second-order differential plot (SODP), e.g., standard descriptors, a summation of the angles between consecutive vectors, a summation of distances to coordinate, a summation of the triangle area using three successive points, etc. [63,64]. The suitable features were selected by utilizing binary particle swarm optimization (PSO) and were fed to support vector machine and k-nearest neighbor (KNN) classifiers for the identification of normal and depressed signals [63]. Further studies are warranted to examine the performance of the aforementioned novel approaches for identifying patterns of EEG signals underlying the neurocognitive endophenotype deficits that occur in schizophrenia. Finally, in a review of fMRI studies regarding the correlations between resting-state network connectivity and attentional performance in schizophrenia, the published studies converged on the notion of attention deficits as related to abnormalities in the default mode network (DMN) [65]. Considering the specificity of neuronal activity assessment using the neuroimaging modality of EEG, which can measure the dynamics of a neural network with excellent temporal resolution and allows for an examination of the whole-brain network functional connectivity in many frequencies, our results await further investigation into whole-brain functional connectivity; this can be investigated using simultaneous EEG-fMRI and a higher spatial sampling of scalp electrodes (i.e., high-density EEG setups) to confirm whether the findings depend on the neuroimaging modality.

## 5. Conclusions

Our study indicates that increased right hemispheric Cu-TTG and Cu-STG resting-state functional connectivity at gamma frequency and increased right hemispheric FG-LG resting-state functional connectivity at beta frequency predicted poorer attentional performance in stabilized patients with schizophrenia. Future studies are needed to confirm whether the networks of these brain regions identified in our study are neural correlates of attention deficits of schizophrenia. If replicated, novel approaches to modulate these networks may yield selective, potent interventions for improving attention deficits in schizophrenia.

## Figures and Tables

**Figure 1 medicina-59-00737-f001:**
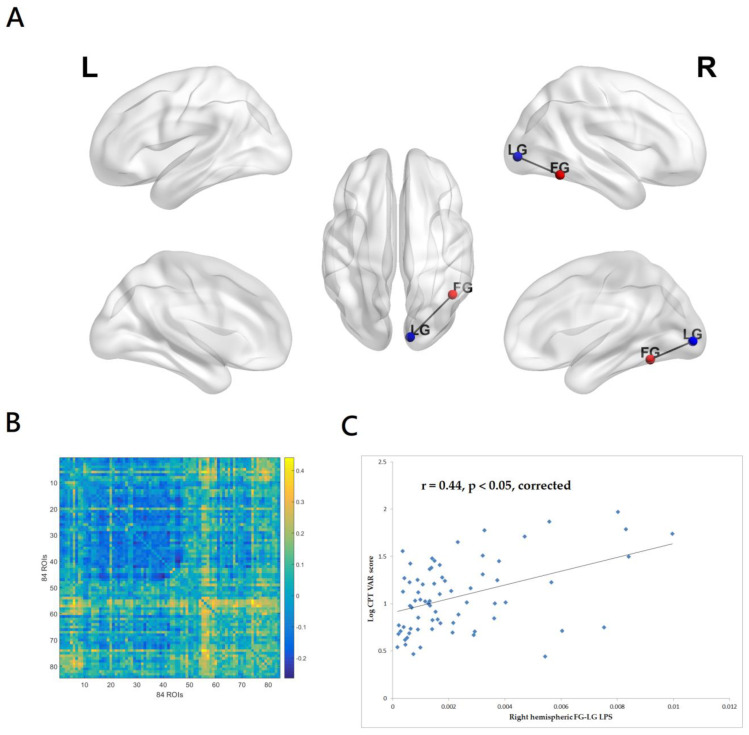
(**A**)The CPT-II VAR score was significantly associated with the beta-band (13–32.5 Hz) LPS between right hemispheric fusiform gyrus (FG) and lingual gyrus (LG). eLORETA and BrainNet Viewer were used to create the figure. (**B**) The matrix of the correlation coefficients for the value of beta-band lagged phase synchronization (LPS) of all pairs from 84 regions of interest. (**C**) Patients with higher CPT-II VAR scores had greater beta-band EEG functional connectivity between right hemispheric FG and LG. The linear regression line was shown to model a linear trend seen in the scatterplot.

**Figure 2 medicina-59-00737-f002:**
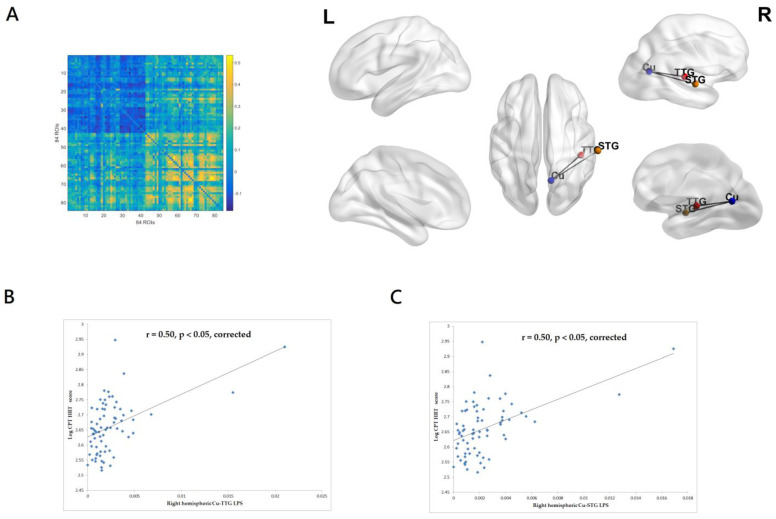
(**A**) Left panel: the matrix of the correlation coefficients for the value of gamma-band (33–45 Hz) lagged phase synchronization (LPS) of all pairs from 84 regions of interest. Right panel: the CPT-II HRT score was significantly associated with the gamma-band LPS between right hemispheric cuneus (Cu) and transverse temporal gyrus (TTG) and between right hemispheric cuneus (Cu) and superior temporal gyrus (STG). eLORETA and BrainNet Viewer were used to create the figure. (**B**) The patients with higher CPT-II HRT scores had greater gamma-band EEG functional connectivity between right hemispheric Cu and TTG and (**C**) between right hemispheric Cu and STG. The linear regression line was shown to model a linear trend seen in the scatterplot.

**Figure 3 medicina-59-00737-f003:**
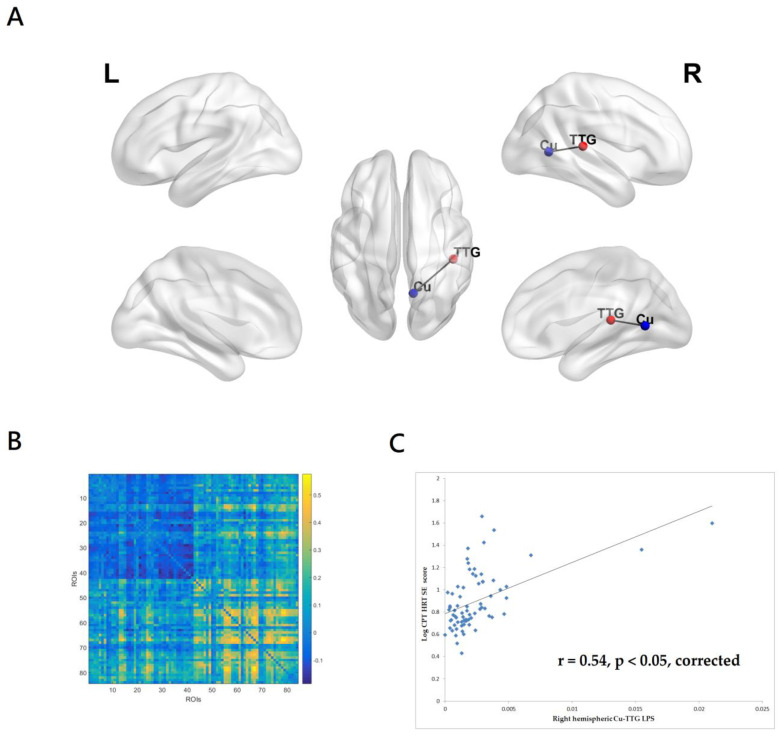
(**A**)The CPT-II HRTSE score was significantly associated with the gamma-band (33–45 Hz) LPS between right hemispheric cuneus (Cu) and transverse temporal gyrus (TTG). eLORETA and BrainNet Viewer were used to create the figure. (**B**) The matrix of the correlation coefficients for the value of gamma-band lagged phase synchronization (LPS) of all pairs from 84 regions of interest. (**C**) The patients with higher CPT-II HRTSE scores had greater gamma-band EEG functional connectivity between right hemispheric Cu and TTG. The linear regression line was shown to model a linear trend seen in the scatterplot.

**Table 1 medicina-59-00737-t001:** Sociodemographic and clinical characteristics of the participants.

	Sample 1 N = 36	Sample 2 N = 36	Total Sample N = 72	*p* Values
Gender (f/m)	15/21	18/18	33/39	0.48
Age, years old	43.33 ± 11.83	42.47 ± 9.96	42.90 ± 10.87	0.74
Years of education, years	13.06 ± 3.00	13.56 ± 3.06	13.31 ± 3.02	0.49
Years since diagnosis, years	18.97 ± 11.56	16.28 ± 10.34	17.44 ± 10.97	0.37
Chlorpromazine equivalent dose, mg/day	593.30 ± 304.15	613.99 ± 410.85	603.64 ± 359.05	0.81
PANSS total score	69.86 ± 9.42	73.28 ± 8.51	71.57 ± 9.08	0.11
PANSS positive subscale	14.36 ± 4.08	16.00 ± 4.46	15.18 ± 4.32	0.11
PANSS negative subscale	18.89 ± 3.21	19.53 ± 3.71	19.21 ± 3.46	0.44
PANSS general subscale	36.61 ± 4.66	37.75 ± 4.63	37.18 ± 4.65	0.30
PSP global scale	55.36 ± 10.80	52.89 ± 10.03	53.13 ± 10.42	0.32
BCIS-R	23.58 ± 4.55	23.81 ± 4.96	23.69 ± 4.73	0.96
BCIS-C	15.00 ± 3.26	16.17 ± 3.00	15.83 ± 3.13	0.37
BCIS R-C index	8.25 ± 4.16	7.64 ± 5.54	7.94 ± 4.87	0.60
CPT-II				
d’	0.61 ± 0.49	0.86 ± 0.59	0.74 ± 0.55	0.05
OM	13.00 ± 21.11	15.67 ± 31.08	14.33 ± 26.41	0.67
COM	17.14 ± 9.69	12.75 ± 8.44	14.94 ± 9.29	0.04
PER	4.17 ± 7.27	3.42 ± 6.67	3.79 ± 6.94	0.65
HRT	455.57 ± 100.16	478.92 ± 105.17	467.25 ± 102.64	0.34
HRTSE	9.65 ± 8.15	9.82 ± 8.19	9.73 ± 8.11	0.93
VAR	17.64 ± 17.46	16.81 ± 18.20	17.23 ± 17.72	0.85
HRTBC	0.01 ± 0.03	0.01 ± 0.03	0.01 ± 0.03	0.29
HRTISIC	0.07 ± 0.04	0.07 ± 0.04	0.06 ± 0.04	0.72
CTT1	60.08 ± 20.93	54.14 ± 25.56	57.11 ± 23.39	0.28
CTT2	105.02 ± 28.10	104.96 ± 39.36	104.99 ± 33.96	0.99
WCST non-perseverative error	21.22 ± 20.45	15.50 ± 16.38	18.36 ± 18.62	0.52
TOL accuracy	4.53 ± 2.13	3.56 ± 2.32	4.04 ± 2.27	0.07
TOL time	234.58 ± 94.30	236.03 ± 91.23	235.31 ± 92.13	0.95
Stroop Interference Test				
Naming interference tendency	0.49 ± 0.45	0.33 ± 0.32	0.41 ± 0.40	0.08
Reading interference tendency	0.28 ± 0.29	0.28 ± 0.24	0.28 ± 0.26	0.84

PANSS, Positive and Negative Syndrome Scale; PSP, Personal and Social Performance scale; BCIS, The Taiwanese version of Beck’s Cognitive Insight Scale; BCIS-R, Self-reflectiveness subscale of BCIS; BCIS-C, Self-certainty subscale of BCIS; CPT-II, Connors’ Continuous Performance Test, 2nd Edition; d’, detection; OM, omission error; COM, commission error; PER, perseveration; HRT, hit reaction time; HRTSE, HRT standard error; VAR, variability; HRTBC, HRT block change; HRTISIC, HRT interstimulus interval change; CTT, Color Trails Test; WCST, Wisconsin Card Sorting Test; TOL, The Tower of London-Drexel University Test 2nd Edition (TOL^DXtm^). Notes: Data are presented as means ± standard deviations unless otherwise stated. *p* values were for the statistical comparisons between sample 1 and sample 2. Pearson chi-square test was used for qualitative variables and Student’s *t*-test or Mann–Whitney test were used for quantitative variables.

**Table 2 medicina-59-00737-t002:** MNI coordinates of the 84 regions of interest (ROIs) from cerebral cortex for the analysis of EEG functional connectivity.

ROI	Structure	x	y	z	ROI	Structure	x	y	z
1	Postcentral Gyrus	−55	−25	50	43	Postcentral Gyrus	55	−25	50
2	Postcentral Gyrus	−45	−30	45	44	Inferior Parietal Lobule	50	−30	45
3	Precentral Gyrus	−35	−25	55	45	Postcentral Gyrus	40	−25	50
4	Precentral Gyrus	−35	−20	50	46	Postcentral Gyrus	35	−25	50
5	Paracentral Lobule	−15	−45	60	47	Paracentral Lobule	15	−45	60
6	Middle Frontal Gyrus	−30	−5	55	48	Middle Frontal Gyrus	30	−5	55
7	Precuneus	−20	−65	50	49	Precuneus	15	−65	50
8	Superior Frontal Gyrus	−20	30	50	50	Superior Frontal Gyrus	20	25	50
9	Middle Frontal Gyrus	−30	30	35	51	Middle Frontal Gyrus	30	30	35
10	Superior Frontal Gyrus	−25	55	5	52	Superior Frontal Gyrus	25	55	5
11	Middle Frontal Gyrus	−20	40	−15	53	Superior Frontal Gyrus	20	45	−20
12	Insula	−40	−10	10	54	Insula	40	−5	10
13	Lingual Gyrus	−10	−90	0	55	Lingual Gyrus	10	−90	0
14	Lingual Gyrus	−15	−85	0	56	Lingual Gyrus	15	−85	0
15	Cuneus	−25	−75	10	57	Cuneus	25	−75	10
16	Fusiform Gyrus	−45	−20	−30	58	Fusiform Gyrus	45	−20	−30
17	Middle Temporal Gyrus	−60	−20	−15	59	Middle Temporal Gyrus	60	−15	−15
18	Superior Temporal Gyrus	−55	−25	5	60	Superior Temporal Gyrus	55	−20	5
19	Posterior Cingulate	−5	−40	25	61	Posterior Cingulate	5	−45	25
20	Cingulate Gyrus	−5	0	35	62	Cingulate Gyrus	5	0	35
21	Medial Frontal Gyrus	−10	20	−15	63	Subcallosal Gyrus	5	15	−15
22	Parahippocampal Gyrus	−20	−35	−5	64	Parahippocampal Gyrus	20	−35	−5
23	Parahippocampal Gyrus	−20	−10	−25	65	Parahippocampal Gyrus	20	−10	−25
24	Posterior Cingulate	−5	−50	5	66	Posterior Cingulate	5	−50	5
25	Posterior Cingulate	−15	−60	5	67	Cuneus	10	−60	5
26	Precuneus	−10	−50	30	68	Precuneus	10	−50	35
27	Anterior Cingulate	−5	30	20	69	Anterior Cingulate	5	30	20
28	Anterior Cingulate	−5	20	20	70	Anterior Cingulate	0	20	20
29	Parahippocampal Gyrus	−15	0	−20	71	Parahippocampal Gyrus	15	0	−20
30	Parahippocampal Gyrus	−20	−25	−20	72	Parahippocampal Gyrus	25	−25	−20
31	Parahippocampal Gyrus	−30	−30	−25	73	Parahippocampal Gyrus	30	−25	−25
32	Fusiform Gyrus	−45	−55	−15	74	Fusiform Gyrus	45	−55	−15
33	Superior Temporal Gyrus	−40	15	−30	75	Superior Temporal Gyrus	40	15	−30
34	Middle Temporal Gyrus	−45	−65	25	76	Middle Temporal Gyrus	45	−65	25
35	Inferior Parietal Lobule	−50	−40	40	77	Inferior Parietal Lobule	50	−45	45
36	Transverse Temporal Gyrus	−45	−30	10	78	Transverse Temporal Gyrus	45	−30	10
37	Superior Temporal Gyrus	−60	−25	10	79	Superior Temporal Gyrus	65	−25	10
38	Transverse Temporal Gyrus	−60	−10	15	80	Transverse Temporal Gyrus	60	−10	15
39	Precentral Gyrus	−50	10	15	81	Precentral Gyrus	55	10	15
40	Inferior Frontal Gyrus	−50	20	15	82	Inferior Frontal Gyrus	50	20	15
41	Middle Frontal Gyrus	−45	35	20	83	Middle Frontal Gyrus	45	35	20
42	Inferior Frontal Gyrus	−30	25	−15	84	Inferior Frontal Gyrus	30	25	−15

## Data Availability

The data presented in this study are available on request from the corresponding author.

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
