# Peer review of "Resting-State EEG Connectivity at High-Frequency Bands and Attentional Performance Dysfunction in Stabilized Schizophrenia Patients"

_medicina, 2023, doi:10.3390/medicina59040737_

Round 1

Reviewer 1 Report

1.      For denoising EEG signals, MSPCA plays vital role, which is a combination of PCA and wavelet. I recommend authors to use MSPCA in discussion,
The details of MSPCA can be found in "Motor imagery BCI classification based on novel two-dimensional modelling in empirical wavelet transform",  “Motor Imagery BCI Classification Based on Multivariate Variational Mode Decomposition”,  “15.Motor Imagery EEG Signals Decoding by Multivariate Empirical Wavelet Transform-Based Framework for Robust Brain–Computer Interface”,  “ Alcoholic EEG signals recognition based on phase space dynamic and geometrical features”

2.      For EEG Signals, convolutional neural networks play vital role, thus details should be included in literature, “Alcoholic EEG signals recognition based on phase space dynamic and geometrical features”.

3.      Graphical features are one of the newest approaches for identifying underlying patterns of EEG signals and details of these methods can significantly increase the readability. Please include discussion from following articles “Depression Detection Based on Geometrical Features Extracted from SODP Shape of EEG Signals and Binary PSO”, “Recognizing seizure using Poincaré plot of EEG signals and graphical features in DWT domain”, 

4.      Please provide the details of future direction and possible solutions to continue this topic.

5.      Finally, I suggest authors to sit with English native speaker to improve the writing of proposed work.

Author Response

Response to reviewer 1

  1. For denoising EEG signals, MSPCA plays vital role, which is a combination of PCA and wavelet. I recommend authors to use MSPCA in discussion,
    The details of MSPCA can be found in "Motor imagery BCI classification based on novel two-dimensional modelling in empirical wavelet transform",  “Motor Imagery BCI Classification Based on Multivariate Variational Mode Decomposition”,  “15.Motor Imagery EEG Signals Decoding by Multivariate Empirical Wavelet Transform-Based Framework for Robust Brain–Computer Interface”,  “ Alcoholic EEG signals recognition based on phase space dynamic and geometrical features”

Response 1

We would like to thank the reviewer for the valuable comment. We add that in line 570-574 “Fourth, the quality of our EEG signals can be further improved by using novel denoising methods, e.g., multiscale principal component analysis (MSPCA) which combines noncentered PCA on approximations and details in the wavelet domain and a 572 final PCA [53-56]. Specifically, empirical wavelet transform decomposes denoised EEG signals into different modes and the two-dimensional modelling of these modes is ap-574 plied to identify the variations of different signals.”.

  1. For EEG Signals, convolutional neural networks play vital role, thus details should be included in literature, “Alcoholic EEG signals recognition based on phase space dynamic and geometrical features”.

Response 2

We would like to thank the reviewer for the valuable comment. We add that in line 575-584 “Next, a single geometrical feature is extracted from 2D modelling of modes and the extracted feature vectors are provided to the machine learning and feedforward neural network (e.g., convolutional neural network, CNN) classifiers and cascade forward neural networks for classification check [56]. Recent research identified the underlying patterns of EEG signals of seizure and a psychiatric phenotype (e.g., depression) by using a novel approach based on geometric features derived from the EEG signal shape of the second-order differential plot (SODP), e.g., standard descriptors, a summation of the angles between consecutive vectors, a summation of distances to coordinate, a summation of the triangle area using three successive points, etc. [57, 58].”

  1. Graphical features are one of the newest approaches for identifying underlying patterns of EEG signals and details of these methods can significantly increase the readability. Please include discussion from following articles “Depression Detection Based on Geometrical Features Extracted from SODP Shape of EEG Signals and Binary PSO”, “Recognizing seizure using Poincaré plot of EEG signals and graphical features in DWT domain”, 

Response 3

We would like to thank the reviewer for the valuable comment. We add that in line 584-586 “The suitable features were selected by utilizing Binary Particle Swarm Optimization (PSO) and were fed to support vector machine and k-nearest 585 neighbor (KNN) classifiers for the identification of normal and depressed signals [57].  Further studies are warranted to examine the performance of the aforementioned novel approaches for identifying patterns of EEG signals underlying the neurocognitive endophenotype deficits that occur in schizophrenia.”

  1. Please provide the details of future direction and possible solutions to continue this topic.

Response 4

We would like to thank the reviewer for the valuable comment. We add that in line 586-589

Further studies are warranted to examine the performance of the aforementioned novel approaches for identifying patterns of EEG signals underlying the neurocognitive endophenotype deficits that occur in schizophrenia.”

  1. Finally, I suggest authors to sit with English native speaker to improve the writing of proposed work.

Response 5

We would like to thank the reviewer for the valuable comment. A native speak help us edit the manuscript.

Reviewer 2 Report

In their manuscript entitled “Resting-state EEG connectivity at high-frequency bands and attentional performance dysfunction in stabilized schizophrenia patients”, the Authors present a study in which they tested possible correlations between the resting-state EEG connectivity and dysfunctions in the attentional performance in patients with schizophrenia.

This is a very interesting study. It seems well designed, well conducted and well written in most of its parts. However, in my opinion, there are some major and minor revisions to be fixed.

Major revisions:

1)      The introduction section lacks of an adequate literature exploration. I think that more references should be described and cited to support the importance of the issue the Authors discussed. Therefore, I suggest Authors to enrich this section with more references.

2)      I suggest Authors to provide a better explanation of the Continuous Performance Test even in the introduction, because the reported information are not sufficient to obtain a clear idea of what the test consists of. Is that a paper and pencil or a computerized test? Is it a self-report test or are the test’s exercises administered by the clinicians?

3)      Linked to the previous point, the Authors started describing the results of some studies applying the Continuous Performance Test on schizophrenia patients (lines 64-75), but again I think that the evidence reported are not enough. Usually, the introduction section of a paper is where the literature and previous data can be reviewed to present and support the main and secondary issues. Therefore, I suggest Authors to complement this part with more citations.  

4)      The final part of the introduction section lacks of a clear statement about the literature gap, that is the issue(s) that, being still unexplored by the literature, justifies the idea of running the study. I suggest Authors to better explain the literature gap. To do so, Authors could use the content they reported at the beginning of the discussion section.

5)      Linked to the previous point, I suggest Authors to better describe their study aim(s), both primary (related to the attentional capacities measure) and secondary (related to the other clinical measures considered).   

6)      Regarding the Table 1, I suggest Authors to run and report the p-value of tests (e.g., t-test) comparing the baseline characteristics of the patients of the two RCTs. This is useful to check for possible baseline confounding factors that could be considered for the main analysis.

Minor revisions:

1)      Regarding the clinical measures, I suggest Authors to report the Cronbach's alpha of the tests used.

Author Response

Response to the reviewer 2

In their manuscript entitled “Resting-state EEG connectivity at high-frequency bands and attentional performance dysfunction in stabilized schizophrenia patients”, the Authors present a study in which they tested possible correlations between the resting-state EEG connectivity and dysfunctions in the attentional performance in patients with schizophrenia.

This is a very interesting study. It seems well designed, well conducted and well written in most of its parts. However, in my opinion, there are some major and minor revisions to be fixed.

Major revisions:

  • The introduction section lacks of an adequate literature exploration. I think that more references should be described and cited to support the importance of the issue the Authors discussed. Therefore, I suggest Authors to enrich this section with more references.

Response 1

We would like to thank the reviewer for the valuable comment. As can be seen in line 81-88, we add more references as follows.

Hirano Y, Uhlhaas PJ. Current findings and perspectives on aberrant neural oscillations in schizophrenia. Psychiatry Clin Neurosci. 2021 Dec;75(12):358-368. doi: 10.1111/pcn.13300. Epub 2021 Oct 29. PMID: 34558155.

Perrottelli A, Giordano GM, Brando F, Giuliani L, Pezzella P, Mucci A, Galderisi S. Unveiling the Associations between EEG Indices and Cognitive Deficits in Schizophrenia-Spectrum Disorders: A Systematic Review. Diagnostics (Basel). 2022 Sep 9;12(9):2193. doi: 10.3390/diagnostics12092193. PMID: 36140594; PMCID: PMC9498272.

Schultheis C, Rosenbrock H, Mack SR, Vinisko R, Schuelert N, Plano A, Süssmuth SD. Quantitative electroencephalography parameters as neurophysiological biomarkers of schizophrenia-related deficits: A Phase II substudy of patients treated with iclepertin (BI 425809). Transl Psychiatry. 2022 Aug 11;12(1):329. doi: 10.1038/s41398-022-02096-5. PMID: 35953474; PMCID: PMC9372178.

Ramsay IS, Pokorny VJ, Lynn PA, Klein SD, Sponheim SR. Limited consistency and strength of neural oscillations during sustained visual attention in schizophrenia. Biol Psychiatry Cogn Neurosci Neuroimaging. 2023 Feb 10:S2451-9022(23)00026-5. doi: 10.1016/j.bpsc.2023.02.001. Epub ahead of print. PMID: 36775194.

  • I suggest Authors to provide a better explanation of the Continuous Performance Test even in the introduction, because the reported information are not sufficient to obtain a clear idea of what the test consists of. Is that a paper and pencil or a computerized test? Is it a self-report test or are the test’s exercises administered by the clinicians?

Response 2

We would like to thank the reviewer for the valuable comment. We add that in line 135-141, “….. Attentional performance are crucial for functional outcomes in patients with schizophrenia. The CPT-II is a commonly used instrument by clinical assessments for detecting attentional deficits in these patients. As far as we know, there is still a lack of research elucidating the relationships between the organizations of resting-state EEG functional connectivity and attentional performance dysfunction during higher-order cognitive processing in the CPT-II. To fill the gap in the available literature, T the primary aim of the current study was to examine…. “

  • Linked to the previous point, the Authors started describing the results of some studies applying the Continuous Performance Test on schizophrenia patients (lines 64-75), but again I think that the evidence reported are not enough. Usually, the introduction section of a paper is where the literature and previous data can be reviewed to present and support the main and secondary issues. Therefore, I suggest Authors to complement this part with more citations.  

Response 3

We would like to thank the reviewer for the valuable comment. We add that in line 81-88 “There is consistent evidence that aberrant neural oscillation at low- and high-frequencies constitute an important aspect of the failure to generate coherent cognition, leading to the characteristic symptoms of cognitive deficits [4, 5]. These deficits 84 are thought to be caused by N-methyl-D-aspartate receptor hypofunction and can be assessed in schizophrenia patient with electroencephalography (EEG) [6]. There’s a growing body of neuroimaging research examining neuroanatomical and neurophysiologicalEEG characterization as a neurophysiological biomarker of attentional dysfunction in 88 schizophrenia and aiming to identify”.

  • The final part of the introduction section lacks of a clear statement about the literature gap, that is the issue(s) that, being still unexplored by the literature, justifies the idea of running the study. I suggest Authors to better explain the literature gap. To do so, Authors could use the content they reported at the beginning of the discussion section.

Response 4

We would like to thank the reviewer for the valuable comment. We add that in line 135-146 “Attentional performance are crucial for functional outcomes in patients with schizophrenia. The CPT-II is a commonly used instrument by clinical assessments for detecting attentional deficits in these patients. As far as we know, there is still a lack of research elucidating the relationships between the organizations of resting-state EEG functional connectivity and attentional performance dysfunction during higher-order cognitive processing in the CPT-II. To fill the gap in the available literature, the primary aim of the current study was to examine he present study aimed to evaluate the correlations between attentional performance assessed by CPT-II and resting-state EEG whole-brain source-based source-level connectivity across a wide range of frequencies in stabilized schizophrenia patients. As secondary aims, we investigated whether EEG connectivity was correlated with the results of clinical measures and other cognitive tests.”

  • Linked to the previous point, I suggest Authors to better describe their study aim(s), both primary (related to the attentional capacities measure) and secondary (related to the other clinical measures considered).   

Response 5

We would like to thank the reviewer for the valuable comment. We add that in line 140-146 “To fill the gap in the available literature, the primary aim of the current study was to examine he present study aimed to evaluate the correlations between attentional performance assessed by CPT-II and resting-state EEG whole-brain source-based source-level connectivity across a wide range of frequencies in stabilized schizophrenia patients. As secondary aims, we investigated whether EEG connectivity was correlated with the results of clinical measures and other cognitive tests.”

6)      Regarding the Table 1, I suggest Authors to run and report the p-value of tests (e.g., t-test) comparing the baseline characteristics of the patients of the two RCTs. This is useful to check for possible baseline confounding factors that could be considered for the main analysis.

Response 6

We would like to thank the reviewer for the valuable comment. We run and report the p-value of tests in Table 1.

Minor revisions:

1)      Regarding the clinical measures, I suggest Authors to report the Cronbach's alpha of the tests used.

Response

We would like to thank the reviewer for the valuable comment. We add that in line 224-228 “.. As reported in our prior work [19, 20], all clinical assessments were done by a well-trained and experienced research psychiatrist (H.C.) and intraclass correlation coefficient for the intrarater repeatability was between excellent and good (0.92 for the PANSS and 0.89 for the PSP). The self-reported Beck Cognitive Insight Scale (BCIS)….”

Reviewer 3 Report

The manuscript is about resting state EEG connectivity in stabilized schizohrenia patients. The reserch question is clinically relevant and the study is well designed. The manuscript is well written. Some points needs addressal:

1- This is a secondary analysis of two different RCTs. The inclusion criteria for both the studies although similar , they differ in few points. This introduces heterogenity in th estudy population . The authors may mention about it in the limitation.

2- Lack of control arm in the design is a majorlacunae. The authors have acknowledged it in the manuscript.

3- Who were the outcome assessors? This should be clarified in the manuscript.

Author Response

Response to the reviewer 3

The manuscript is about resting state EEG connectivity in stabilized schizohrenia patients. The reserch question is clinically relevant and the study is well designed. The manuscript is well written. Some points needs addressal:

  • This is a secondary analysis of two different RCTs. The inclusion criteria for both the studies although similar , they differ in few points. This introduces heterogenity in th estudy population . The authors may mention about it in the limitation.

Response 1

We would like to thank the reviewer for the valuable comment. We add that in line 560-561, “…..because this study did not have a control group of healthy individuals and because there was some heterogeneity among the two samples analyzed. Further studies including both homogeneous patients and healthy…. “

  • Lack of control arm in the design is a majorlacunae. The authors have acknowledged it in the manuscript.

Response 2

We would like to thank the reviewer for the valuable comment.

3- Who were the outcome assessors? This should be clarified in the manuscript.

Response 3

We would like to thank the reviewer for the valuable comment. We add that in line 224-228 “.. As reported in our prior work [19, 20], all clinical assessments were done by a well-trained and experienced research psychiatrist (H.C.) and intraclass correlation coefficient for the intrarater repeatability was between excellent and good (0.92 for the PANSS and 0.89 for the PSP). The self-reported Beck Cognitive Insight Scale (BCIS)….”

Round 2

Reviewer 1 Report

The authors revised article very carefully. I have no further questions.

Author Response

We would like to thank the reviewer's comment. 

Reviewer 2 Report

In their manuscript entitled “Resting-state EEG connectivity at high-frequency bands and attentional performance dysfunction in stabilized schizophrenia patients”, the Authors present a study in which they tested possibile correlations between the resting-state EEG connectivity and dysfunctions in the attentional performance in patients with schizophrenia. This is a very interesting and tamily study. It seems well designed, well conducted and well written in most of its parts. I really appreciated the Authors’ efforts to address the revisions. However, in my opinion, there are still a couple of major revisions and a minor revision to be fixed.

Major revisions:

- I appreciated the Authors’ revisions, but I think that the introduction section can be still improved in terms of reported data and references. Indeed, in my opinion, the introduction still lacks an adequate literature exploration. I think that more references should be descripted and cited to support the importance of the issue the Authors discussed. Therefore, I suggest Authors to enrich this section with more references.

- Linked to the previous point, I appreciated that Authors added more detailed information about the Continuous Performance Test. However, regarding the section describing the results of some studies applying the Continuous Performance Test on schizophrenia patients, I still think that the evidence reported are not enough. Therefore, I suggest Authors to complement this part with more citations.  

Minor revisions:

- Regarding the Table 1, I appreciated that Authors added the p-values. However, I suggest Authors to specify, for instance by using a Table note, which test(s) they ran to compare the two samples.  

Author Response

In their manuscript entitled “Resting-state EEG connectivity at high-frequency bands and attentional performance dysfunction in stabilized schizophrenia patients”, the Authors present a study in which they tested possibile correlations between the resting-state EEG connectivity and dysfunctions in the attentional performance in patients with schizophrenia. This is a very interesting and tamily study. It seems well designed, well conducted and well written in most of its parts. I really appreciated the Authors’ efforts to address the revisions. However, in my opinion, there are still a couple of major revisions and a minor revision to be fixed.

Major revisions:

- I appreciated the Authors’ revisions, but I think that the introduction section can be still improved in terms of reported data and references. Indeed, in my opinion, the introduction still lacks an adequate literature exploration. I think that more references should be descripted and cited to support the importance of the issue the Authors discussed. Therefore, I suggest Authors to enrich this section with more references.

Response 1

We would like to thank the reviewer for the valuable comment. We add that in line 63-86. References 2-9 were added.

- Linked to the previous point, I appreciated that Authors added more detailed information about the Continuous Performance Test. However, regarding the section describing the results of some studies applying the Continuous Performance Test on schizophrenia patients, I still think that the evidence reported are not enough. Therefore, I suggest Authors to complement this part with more citations. 

Response 2

We would like to thank the reviewer for the valuable comment. We add that in line 63-86. References 2-9 were added.

Minor revisions:

- Regarding the Table 1, I appreciated that Authors added the p-values. However, I suggest Authors to specify, for instance by using a Table note, which test(s) they ran to compare the two samples. 

Response 3

We would like to thank the reviewer for the valuable comment. We add that in line 184-186